# Nostril-specific and structure-based olfactory learning of chiral discrimination in human adults

**Guo Feng**[1,2,3], **Wen Zhou**[1,2]*

[1]State Key Laboratory of Brain and Cognitive Science, CAS Center for Excellence in Brain Science and Intelligence Technology, Institute of Psychology, Chinese Academy of Sciences, Beijing, China; [2]Department of Psychology, University of Chinese Academy of Sciences, Beijing, China; [3]Psychological Research and Counseling Center, Southwest Jiaotong University, Chengdu, China

**Abstract** Practice makes perfect. In human olfaction, such plasticity is generally assumed to occur at the level of cortical synthetic processing that shares information from both nostrils. Here we present findings that challenge this view. In two experiments, we trained human adults unirhinally for the discrimination between odor enantiomers over a course of about 10 to 11 days. Results showed that training-induced perceptual gain was restricted to the trained nostril yet partially generalized to untrained odor enantiomers in a structure- rather than quality- based manner. In other words, learning enhanced the differentiation of chirality (molecular configuration) as opposed to overall odor quality (odor object) per se. These findings argue that, unlike earlier beliefs, one nostril does not readily know what the other learns. Moreover, the initial analytical processing of the structural features of uninarial olfactory input remains plastic in human adults.
DOI: https://doi.org/10.7554/eLife.41296.001

*For correspondence:
zhouw@psych.ac.cn

**Competing interests:** The authors declare that no competing interests exist.

## Introduction

Asymmetric carbon atoms, those with four chemically distinct substituents, cause chirality in molecules, that is, the possibility of two nonidentical mirror image forms called enantiomers. Whereas the different enantiomers of a chiral molecule have almost identical physical and chemical properties, they usually have different biological properties and can smell differently to the human nose (*Friedman and Miller, 1971*; *Russell and Hills, 1971*). After all, olfactory receptors are proteins that use only one enantiomeric form (the L form) of the amino-acid building blocks (*Mombaerts, 1999*). The olfactory ability of chiral discrimination has genetic basis (*Polak et al., 1989*; *Saito et al., 2009*) but can also be acquired through learning by human adults (*Li et al., 2008*), which highlights the plasticity of the adult olfactory system. The mechanism underlying such plasticity is not yet fully understood. In vision, the specificity of perceptual learning has long provided important clues to the locus of cortical plasticity (*Gilbert, 1994*). For instance, learning of texture discrimination is specific for retinal input with little interocular transfer (*Karni and Sagi, 1991*), and is subserved by dynamic changes of activities in a subregion of the primary visual cortex (V1) corresponding to the trained visual field quadrant (*Yotsumoto et al., 2008*). In olfaction, an ancient sense that resides in the allocortex, the specificity of perceptual learning has not been systematically characterized. An earlier study tested individuals who could not smell androstenone and found that repeatedly exposing one nostril of these individuals to androstenone improved the androstenone-detection accuracy in both the exposed nostril and the unexposed nostril to the same extent (*Mainland et al., 2002*). The consensus seems to be that olfactory learning takes place at the level of cortical synthetic processing

**eLife digest** Although we may only become consciously aware of our sense of smell when we encounter something pungent, it can greatly influence our quality of life. Smells are processed by our olfactory system, a collection of receptors and nerve cells in the nose and brain.

Odor molecules activate the olfactory system when they bind to receptors in the nostrils. These molecules can have a wide range of chemical and physical properties. However, some odor molecules are mirror images of each other. These variants are known as enantiomers. Some people can naturally smell the difference between enantiomers; others can be taught how to tell them apart. Studying this training process could help us to understand how the olfactory system adapts to new circumstances.

Feng and Zhou trained volunteers to distinguish between odor enantiomers – but only using one nostril. After training, the volunteers were better able to tell the difference between different enantiomers – even for certain scents they had not been trained to discriminate – when sniffing through the trained nostril. However, they got no better at distinguishing between enantiomers when they sniffed them using the other nostril.

Overall, the results reported by Feng and Zhou confirm that human adults can learn how to process the structural features of odorants. However, in contrast to earlier beliefs, they also suggest that one nostril does not readily know what the other learns. This new understanding of how the olfactory system adapts could ultimately help us to develop therapies to restore a lost sense of smell.

DOI: https://doi.org/10.7554/eLife.41296.002

that shares information from both nostrils, rather than initial analytical processing of chemical features (*Mainland et al., 2002*; *Wilson and Stevenson, 2003*).

Unlike the detection of androstenone, which could involve perceptual criterion (*Bremner et al., 2003*), the discrimination of odor enantiomers ultimately draws upon the enantioselectivity of olfactory receptors. The specificity/generalization of chiral discrimination learning thus offers a unique window into where the plasticity underlying olfactory learning firstly occurs. In two experiments, we trained participants unirhinally for the discrimination between odor enantiomers (procedure illustrated in *Figure 1a*) and assessed the extent to which the learning effect transferred to the untrained nostril or an untrained enantiomer pair that is structurally distinct from (Experiments 1 and 2) or similar to (Experiment 2) the enantiomer pair used for training.

## Results

The olfactory stimuli in Experiment 1 consisted of two structurally distinct pairs of enantiomers, the enantiomers of α-pinene and those of 2-butanol (*Figure 1b*). Each served as the training pair for half of the participants and the control pair for the other half. Chiral discrimination was assessed with a triangular unirhinal odor discrimination task. In each trial of the task, participants sampled three bottles (two containing one odorant, the other containing its chiral counterpart), one at a time, by sniffing with one nostril (the other nostril was pinched firmly shut), and then selected the odd one out. At baseline (pretest), discrimination accuracies were at chance (overall accuracy = 0.37 vs. chance = 0.33; $t_{11} = 1.77$, $p = 0.10$; binomial test, $p > 0.3$) irrespective of the enantiomer pair presented (α-pinene vs. 2-butanol, $t_{11} = 1.07$, $p = 0.31$) or the nostril of presentation (left vs. right, $t_{11} = -0.64$, $p = 0.54$). Training began the day after pretest and comprised sessions spaced 1 day apart (*Figure 1a*), where participants completed 12 trials of the unirhinal odor discrimination task and received immediate feedback about the correct choice after each trial. Such reinforcement signals have been shown to guide plasticity and induce perceptual learning even in the absence of awareness of the stimulus presentation (*Roelfsema et al., 2010*). Only the training pair of enantiomers was presented, always to the same nostril (either the left or the right nostril) for a given participant. Training concluded when participants reached a criterion of 9 correct choices out of 12 trials (75%) on two consecutive sessions. A posttest was conducted an hour later, which, like the pretest, involved both the training and the control enantiomer pairs presented to each nostril, without feedback.

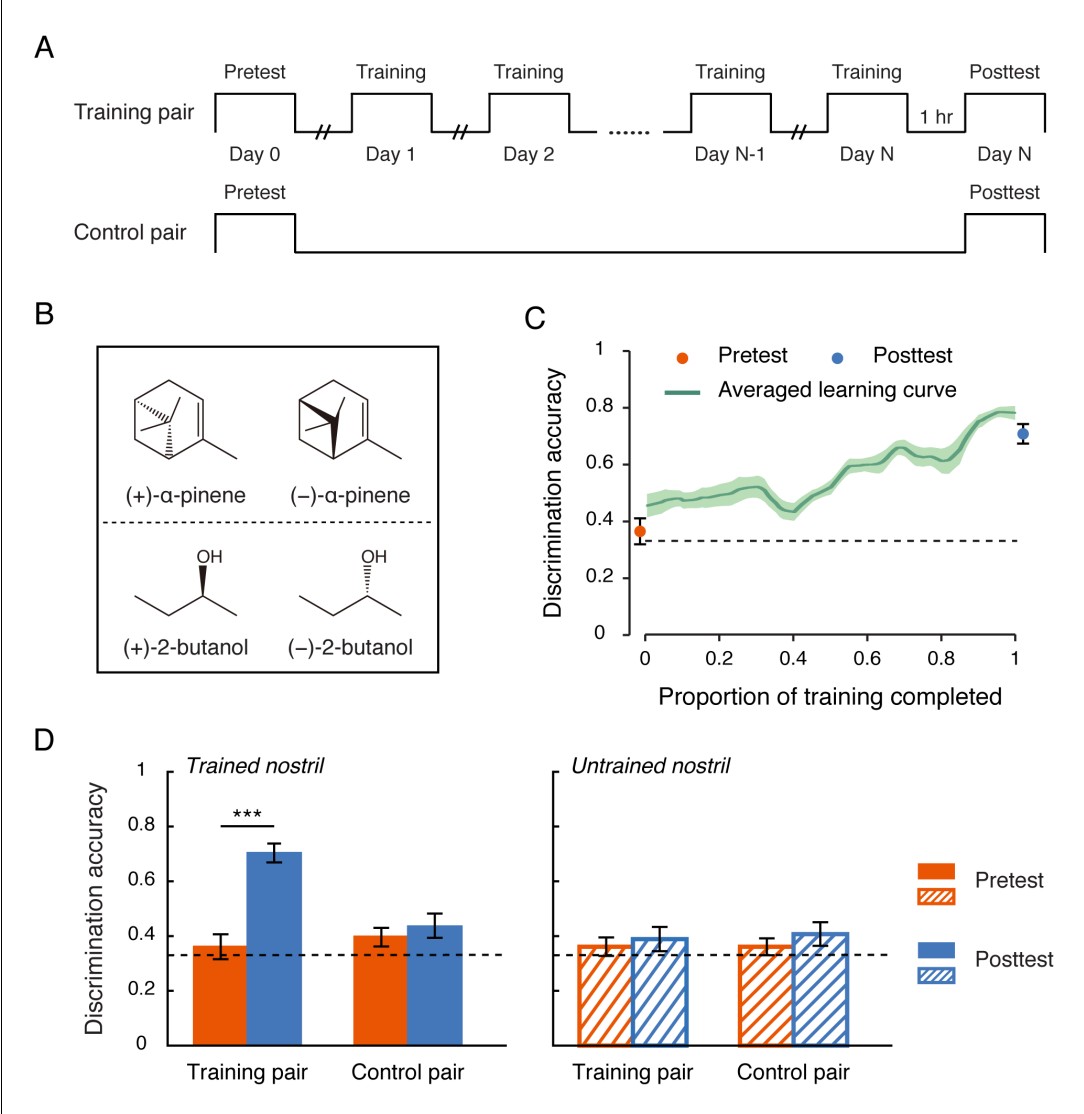

**Figure 1.** Nostril-specific olfactory learning of chiral discrimination. (a) Schematic illustration of the experimental procedure, which comprised of three phases: pretest, training and posttest. During the training phase, participants were trained unirhinally for chiral discrimination with a pair of odor enantiomers. (b) Chemical structures of the enantiomers of α-pinene and 2-butanol. Each enantiomer pair served as the training pair for half of the participants in Experiment 1 and the control pair for the other half. (c) Improvements in chiral discrimination over the course of training (green curve). Data points were linearly interpolated and averaged across participants. Shaded area represents SEMs. Dots mark the mean discrimination accuracies at pretest and posttest for the training pair of enantiomers presented to the trained nostril. (d) Chiral discrimination accuracies at pretest and posttest for the training pair and the control pair of enantiomers presented to the trained vs. untrained nostril. Dashed lines: chance level (0.33); error bars: SEMs; ***p < 0.001.

DOI: https://doi.org/10.7554/eLife.41296.003

The following source data and figure supplement are available for figure 1:

**Source data 1.** Participants' chiral discrimination accuracies at pretest and posttest in Experiment 1.

DOI: https://doi.org/10.7554/eLife.41296.005

**Figure supplement 1.** Chiral discrimination accuracies at pretest and posttest for each pair of odor enantiomers presented to the trained vs. untrained nostril for individual participants in Experiment 1.

DOI: https://doi.org/10.7554/eLife.41296.004

Training lasted 12.1 days on average, during which period the participants' discrimination accuracies showed a continuous improvement from chance level (*Figure 1c*). It is worth noting that the increments in performance were long-term and referred to learning retained from one daily session to the next. Critically, this improvement did not transfer to the untrained nostril nor to the structurally distinct control enantiomer pair. A direct comparison of the participants' performances at pretest and posttest (*Figure 1d*) showed that training doubled the chiral discrimination accuracy for the training pair of enantiomers presented to the trained nostril (from 0.36 to 0.70, $t_{11}$ = 6.37, $p < 0.0001$, Cohen's d = 1.84), but failed to affect the discrimination accuracy for the training pair presented to the untrained nostril ($t_{11}$ = 0.52, $p$ = 0.61), or the control pair presented to the trained ($t_{11}$ = 0.60, $p$ = 0.56) or untrained nostril ($t_{11}$ = 0.84, $p$ = 0.42). Meanwhile, several participants noted at posttest that the trained enantiomers smelled much weaker when presented to the trained as opposed to the untrained nostril.

These results presented a sharp contrast to the internarial transfer of learning of androstenone detection (*Mainland et al., 2002*), and pointed to plastic changes at a level in the olfactory system where uninarial information is still retained. Considering that olfactory perception begins with systematic mappings of chemical structural features (*Khan et al., 2008*) yet relies heavily on memory (*Wilson and Stevenson, 2003*), a natural question that followed was whether such plasticity reflected enhanced differentiation of chirality (molecular configuration) or enhanced differentiation of overall odor quality (odor object) per se (*Rabin, 1988*) with respect to the trained enantiomers. In the former but not the latter case, we reasoned that learning would generalize, to a certain extent, to an enantiomer pair that is perceptually different from but structurally similar to the trained pair in terms of the substituents attached to the chiral center.

To tease apart the above alternatives, we adopted two structurally similar pairs of enantiomers in Experiment 2, the enantiomers of carvone and those of limonene (*Figure 2a*). They share an isopropenyl group at the chiral center and a methyl group at the para-position, that is, at the opposite side of the ring structure relative to the chiral carbon atom bearing the isopropenyl group, and differ only by a carbonyl group. We also included the enantiomers of α-pinene (*Figure 1b*) as a structurally dissimilar control. Carvone, limonene and α-pinene are all found naturally in plants yet smell quite different from one another. Since many volunteers readily distinguished the (+) and (-) forms of carvone and limonene (*Laska and Teubner, 1999*), we screened participants for the inability to do so and were able to recruit 12 non-discriminators (out of 46 tested, 6 females). As a group, they showed chance-level discrimination accuracies at baseline (pretest, overall accuracy = 0.37 vs. chance = 0.33; $t_{11}$ = 1.72, $p$ = 0.11; binomial test, $p > 0.3$) regardless of the enantiomer pair presented (carvone vs. limonene vs. α-pinene, $F_{2, 22}$ = 1.34, $p$ = 0.28) or the nostril of presentation (left vs. right, $F_{1, 11}$ = 0.37, $p$ = 0.55). Half of them were subsequently trained unirhinally to discriminate the enantiomers of carvone, and the other half the enantiomers of limonene, following identical procedures as in Experiment 1.

Training lasted for 9.1 days on average and yielded a similar learning curve to that in Experiment 1 (*Figure 2b*). The improvement in chiral discrimination was again largely confined to the trained nostril. At posttest relative to pretest, the participants showed a substantially enhanced ability to discriminate the training pair of enantiomers presented to the trained (from 0.37 to 0.72, $t_{11}$ = 10.65, $p < 0.0001$, Cohen's d = 3.08), but not the untrained nostril ($t_{11}$ = 2.11, $p$ = 0.059), with a significant difference between the two nostrils ($t_{11}$ = 6.05, $p < 0.0001$, Cohen's d = 1.75, *Figure 2c*). In the meantime, they also experienced a reduction in perceived intensity when these enantiomers were presented to the trained ($t_{11}$ = −4.69, $p$ = 0.001, Cohen's d = −1.35), but not the untrained ($t_{11}$ = 0, $p > 0.9$), nostril, consistent with the participants' reports in Experiment 1 (*Figure 2d*).

Importantly, within the trained nostril, a structure-based generalization was evident (*Figure 2c*): Individuals trained to discriminate the enantiomers of carvone also demonstrated improved discrimination for the enantiomers of limonene (Wilcoxon signed rank test, $p$ = 0.026), but not for the enantiomers of α-pinene ($p$ = 0.32), and vice versa (carvone: $p$ = 0.039; α-pinene: $p$ = 0.60). Overall, discrimination accuracies for the untrained but structurally similar enantiomer pair increased by 25.9% ($t_{11}$ = 7.00, $p < 0.0001$, Cohen's d = 2.02) — to a lesser extent as compared with the trained pair (35.2%, $t_{11}$ = −2.59, $p$ = 0.025, Cohen's d = −0.75) — when it was presented to the trained nostril, and stayed unchanged when it was presented to the untrained nostril ($t_{11}$ = 1.17, $p$ = 0.27). Those for the structurally dissimilar enantiomer pair (i.e. the enantiomers of α-pinene) remained unaltered regardless of the nostril of presentation (trained: $t_{11}$ = 0.58, $p$ = 0.57; untrained: $t_{11}$ = 0.90,

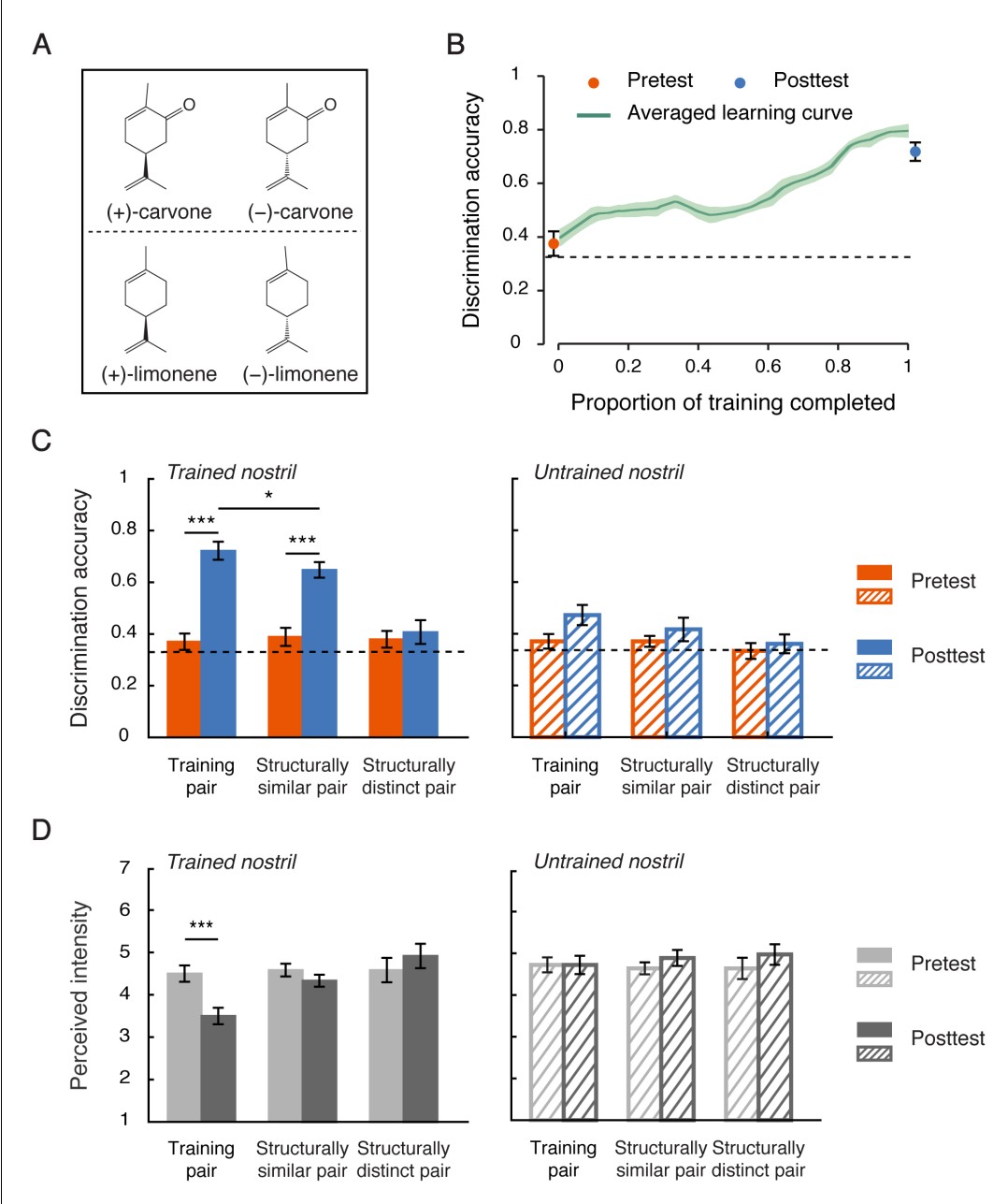

**Figure 2.** Structure-based generalization of chiral discrimination learning. (a) Chemical structures of the enantiomers of carvone and limonene, differing only by a carbonyl group. Each enantiomer pair was used for training for half of the participants in Experiment 2. (b) Improvements in chiral discrimination over the course of training (green curve). Data points were linearly interpolated and averaged across participants. Shaded area represents SEMs. Dots mark the mean discrimination accuracies at pretest and posttest for the training pair of enantiomers presented to the trained nostril. (c) Chiral discrimination accuracies at pretest and posttest for the training pair (carvone or limonene) and the non-training pairs of enantiomers, one structurally similar to the training pair (limonene or carvone) and one structurally distinct ($\alpha$-pinene), presented to the trained vs. untrained nostril. (d) Intensity ratings at pretest and posttest for the three pairs of enantiomers (ratings averaged between the two enantiomers in each pair) presented to the trained vs. untrained nostril. Dashed lines: chance level (0.33); error bars: SEMs; *$p < 0.05$; ***$p < 0.001$.
DOI: https://doi.org/10.7554/eLife.41296.006

The following source data and figure supplement are available for figure 2:

**Source data 1.** Participants' chiral discrimination accuracies and intensity ratings at pretest and posttest in Experiment 2.
DOI: https://doi.org/10.7554/eLife.41296.008

**Figure supplement 1.** Chiral discrimination accuracies at pretest and posttest for each pair of odor enantiomers presented to the trained vs. untrained nostril for individual participants in Experiment 2.

*Figure 2 continued on next page*

*Figure 2 continued*

DOI: https://doi.org/10.7554/eLife.41296.007

p = 0.39). There was no significant change in perceived intensity for the untrained enantiomers, structurally similar or dissimilar, from pretest to posttest irrespective of which nostril they were presented to (ps > 0.1, *Figure 2d*). In other words, the acquisition of chiral discrimination was not necessarily accompanied by a change in the perceived saliency (intensity) of the enantiomers' smells. It was also unlikely that overall perceptual quality contributed to the observed generalization, as the participants rated the racemic mixtures of carvone, limonene and α-pinene as smelling distinctively different from one another (mean similarity rating <20 for any two of the three compounds, on a 100-unit visual analogue scale with 100 denoting extremely similar) and distinguished them with ceiling-level accuracy (>0.9 for any two of them vs. chance = 0.33, triangular binarial odor discrimination test). Taken together, these results suggested that the plasticity underpinning the acquisition of chiral discrimination originates in early olfactory regions that analyze the structural features (i.e. chirality) of uninarial olfactory input. We note that they did not fully exclude the possibility that the participants performed the chiral discriminations based on a learned sub-quality of the odors.

## Discussion

Chirality represents a simple chemical feature of odorants. The specificity to nostril of training and to the structure (rather than overall quality) of the trained compound in the learning of chiral discrimination points to the involvement of early stages in olfactory processing, where uninarial information is retained and structural features are extracted and mapped. In the olfactory system, inputs from the two nostrils remain largely separated up to the primary olfactory cortex including the piriform cortex, the largest recipient of bulbar projections that is heavily implicated in odor object perception (*Carmichael et al., 1994*; *de Olmos et al., 1978*; *Gottfried, 2010*). But in contrast to the chemotopy seen in the olfactory bulb (*Khan et al., 2008*; but see *Soucy et al., 2009*), neurons in the piriform cortex exhibit no correspondence between chemical receptive field and position, show minimal cross-habituation between odorants differing by as few as two carbons, and have been proposed to code synthetic odorant identity and odor quality (*Kadohisa and Wilson, 2006*; *Stettler and Axel, 2009*; *Wilson, 2000*). Other primary olfactory regions have not been shown to be heavily engaged in odor discrimination. In addition, axonal projections from the olfactory bulb to the anterior olfactory nucleus pars principalis, the amygdala and the olfactory tubercle exhibit only a crude set of topographical relationships (*Friedrich, 2011*; *Giessel and Datta, 2014*; *Miyamichi et al., 2011*). We therefore infer that an initial locus for plasticity underlying the learning of chiral discrimination likely resides in or upstream of the olfactory bulb. Experience-dependent modulations of olfactory receptor expressions in the epithelium (*Tan et al., 2015*), odor-induced oscillatory responses in the bulb (*Kay et al., 2009*) and/or tuning of the receptive fields of mitral-tufted cells (*Fletcher and Wilson, 2003*; *Kato et al., 2012*) could in turn be utilized by downstream regions like the posterior piriform to facilitate perceptual discrimination between odor enantiomers (*Li et al., 2008*). In mice, adult neurogenesis is found to reinforce functional inhibition in the olfactory bulb and critically mediate olfactory perceptual learning (*Moreno et al., 2009*), but it is highly controversial whether adult neurogenesis exists in human brains (*Bergmann et al., 2012*; *Curtis et al., 2007*; *Sorrells et al., 2018*). The anterior olfactory nucleus pars externa precisely links mirror-symmetric isofunctional mitral-tufted cells between the olfactory bulbs (*Grobman et al., 2018*; *Yan et al., 2008*), which in theory could provide bilateral access to unilaterally stored olfactory associations (*Kucharski and Hall, 1987*). However, as only a subset of mitral-tufted cells are interconnected and the connections are weak (*Grobman et al., 2018*), the activities of the mirror-symmetric mitral-tufted cells, if any, are likely insufficient to enable discrimination between the trained enantiomers presented to the untrained nostril, resulting in the observed nostril-specific gain in chiral discrimination. The exact neural substrates subserving human olfactory learning await further investigation.

Different theories of perceptual learning have been developed mainly based on studies of visual perceptual learning. They assume that practice improves discrimination by enhancing the signal (*Gold et al., 1999*; *Seitz and Dinse, 2007*), filtering/reducing external and internal noise (*Dosher and Lu, 1998*) or inducing a top-down cascade of weight retuning (*Ahissar and Hochstein,*

*2004*). As perceptual training involves exposure to stimuli and consequently adaptation, the impact of adaptation on task performance has been hard to discern (*Ahissar and Hochstein, 2004*; but see *Harris et al., 2012*). Participants in the current study consistently showed nostril-specific adaptation to the enantiomers used for training as well as nostril-specific learning of chiral discrimination. In other words, training-induced perceptual gain was not accompanied by an overall enhancement of the perceptual salience (intensity) of the trained odor enantiomers. Neither did it require perceptual adaptation, as learning generalized in a structure-based manner to an untrained enantiomer pair for which the participants reported no change in perceived intensity (Experiment 2). As such, our results seem more consistent with the reverse hierarchy theory (*Ahissar and Hochstein, 2004*), which asserts that learning is a gradual top-down-guided (e.g., feedback about the correct choice) increase in usability of first high- then lower-level task-relevant information and that this process is subserved by a cascade of top-to-bottom level modifications that enhance task-relevant information (e.g., configurations of the substituents attached to the chiral center). We note that this framework also nicely reconciles our results with the internarial transfer of learning of androstenone detection observed earlier (*Mainland et al., 2002*). Supposing that learning proceeds as a countercurrent along the olfactory processing hierarchy, the detection of androstenone could be achieved by heightened attention or sensitization to its olfactory quality (amplification of a weak signal which lowers detection threshold) and hence possibly engages a higher generalizing level, whereas the learning of chiral discrimination likely relies more on lower-level inputs and is thus more specific.

On a separate note, a previous study by researchers in Germany tested the olfactory discrimination ability of 20 participants for 10 pairs of enantiomers (*Laska and Teubner, 1999*) and found that they significantly discriminated the optical isomers of α-pinene, carvone and limonene (mean accuracies ≈ 80% for each enantiomer pair vs. chance = 33%) despite marked individual differences in discrimination performance. In our testing, we also noticed that many individuals readily distinguished the (+) and (-) forms of carvone and limonene and had to screen participants for the inability to do so in Experiment 2. However, we did not encounter any individual who could reliably tell apart the enantiomers of α-pinene without training. It is unclear whether this interesting discrepancy was due to genetic or environmental differences between Chinese and German participants. But it indicates that structural properties of a chiral compound cannot fully predict whether the enantiomers smell differently to a human recipient.

To summarize, the current study suggests that human olfactory perception can partially be a learned trait. Learning can take place at a very early stage of olfactory processing, where the structural features of odorants are analyzed and extracted. Put differently, early analytical processing of the chemical features of unirhinal input is both plastic (experience-dependent) and behaviorally relevant in human adults. Unlike earlier beliefs (*Mainland et al., 2002*), one nostril does not always know what the other learns.

## Materials and methods

### Participants

A total of 24 healthy non-smokers completed the training for chiral discrimination, which lasted for 10 to 11 days on average: 12 (6 males, mean age ±SD = 24.8 ± 2.1 years) in Experiment 1 and another 12 (6 males, 21.3 ± 2.0 years; screened from 46 volunteers) in Experiment 2. The sample size was motivated by those used in previous studies (*Karni and Sagi, 1991*; *Mainland et al., 2002*). All participants reported to have a normal sense of smell and no respiratory allergy or upper respiratory infection at the time of testing. Written informed consent and consent to publish was obtained from participants in accordance with ethical standards of the Declaration of Helsinki (1964). The study was approved by the Institutional Review Board at Institute of Psychology, Chinese Academy of Sciences.

### Olfactory stimuli

The olfactory stimuli were presented in identical 280 ml narrow-mouthed glass bottles. They consisted of the enantiomers of α-pinene (1% v/v in propylene glycol) and 2-butanol (1% v/v in propylene glycol) in Experiment 1 (*Figure 1b*), and the enantiomers of carvone (0.5% v/v in propylene glycol), limonene (0.5% v/v in propylene glycol) and α-pinene (1% v/v in propylene glycol), as well as

their racemic mixtures (0.5%, 0.5% and 1% v/v in propylene glycol, respectively), in Experiment 2 (*Figure 2a*). Each bottle contained 10 ml clear liquid and was connected with either one (for those containing an odor enantiomer) or two (for those containing a racemic mixture) Teflon nosepieces. Participants were instructed to sample the olfactory stimuli by inhaling through the nosepieces and exhaling through the mouth.

At the concentrations used, the odor enantiomers were clearly detectable yet did not elicit a significant trigeminal response, as assessed in an independent panel of 20 participants (10 males, 22.6 ± 1.7 years), who failed to localize the side of uninarial stimulation for all of them in a lateralization test (*Wysocki et al., 2003*) (32 trials per participant, mean accuracy for each enantiomer <0.52 vs. chance = 0.5, ps >0.7).

## Procedure

Experiment 1 comprised three phases: pretest, training and posttest (*Figure 1a*). At pretest (day 0) and posttest (day N), the participants performed 36 trials of a triangular unirhinal odor discrimination task. In each trial of the task, they were blindfolded and were instructed to close either the left or the right nostril with the index finger. The open nostril was subsequently presented with three bottles, two containing one odorant, the other containing its chiral counterpart, one at a time in random order, and the participants were asked to select the odd one out. Specifically, the nosepieces connected with the bottles were positioned into the open nostril by the experimenter, one at a time, without the participants holding or touching the bottles. There were 9 trials per enantiomer pair per nostril, with a break of at least 30 s in between two trials. The order of the enantiomer pairs was counterbalanced across participants (i.e., half of the participants first performed the task with the enantiomers of α-pinene, followed by the enantiomers of 2-butanol; the other half did the reverse). For each participant, the nostril of presentation (open nostril) was alternated in consecutive trials. No feedback was provided.

Training began the day after pretest (day 1) and consisted of sessions spaced 1 day apart (day 1 to day N). In each session, the participants, blindfolded, completed 12 trials of the triangular unirhinal chiral discrimination task and received immediate feedback about the correct choice. Half of the participants were randomly assigned to be trained with the enantiomers of α-pinene and the other half the enantiomers of 2-butanol. The training pair of enantiomers was always presented to the left nostril for 5 of the participants (randomly assigned, 2 trained with the enantiomers of α-pinene) and to the right nostril for the other 7 (4 trained with the enantiomers of α-pinene). We limited the number of trials in each session to reduce olfactory fatigue, as participants in our pilot testing reported that they could only smell a faint odor after about 12 trials, which were at least 36 repetitive exposures to the (+) and (-) forms of the same chiral compound presented in the same nostril. Training concluded when the participants reached a criterion of 9/12 correct choices on two consecutive sessions (day N-1 and day N). The posttest was conducted an hour later (day N). The same bottles used for the trained nostril were also used for the untrained nostril. During the course of the experiment, the odor solution in each bottle was replaced with a freshly prepared solution of the same compound every two days.

Experiment 2 followed similar procedures to those described above, except for the followings: Participants were screened for the inability to distinguish the (+) and (-) forms of carvone and limonene (accuracy <0.56 for each enantiomer pair vs. chance = 0.33). At pretest (day 0) and posttest (day N), each participant completed 54 trials of the triangular unirhinal odor discrimination task (9 trials per enantiomer pair per nostril, 3 pairs of enantiomers). Prior to the odor discrimination task, they rated the intensity of each odor enantiomer, presented monorhinally, on a 7-point Likert scale, with 7 signifying very strong. In the training phase (day 1 to day N), half of the participants were trained with the enantiomers of carvone and the other half the enantiomers of limonene. The training pair of enantiomers was always presented to the left nostril for half of the participants (3 trained with the enantiomers of carvone) and to the right nostril for the other half (3 trained with the enantiomers of carvone). After the posttest, 10 of the participants sampled the racemic mixtures of carvone, limonene and α-pinene, presented birhinally, and provided ratings for the perceived similarities between every two of them on a 100-unit visual analogue scale, with 100 representing extremely similar. They also completed a triangular odor discrimination task using these three racemic mixtures. In each trial of the task, they were blindfolded and were presented birhinally with three bottles, two containing one racemic mixture, the other containing a different racemic mixture, one

at a time in random order, and were asked to select the odd one out. There were 12 trials, 4 trials for every pairwise combination, with a break of at least 30 s in between two trials. No feedback was provided.

## Analysis

For each of Experiments 1 and 2, the participants' chiral discrimination accuracies at pretest were analyzed in a repeated measures ANOVA, using enantiomer pair (Experiment 1: α-pinene vs. 2-butanol; Experiment 2: carvone vs. limonene vs. α-pinene) and nostril of presentation (left vs. right) as the within-subject factors. The overall accuracies were subsequently compared against chance (1/3) in a one-sample t test and a binomial test (averaged number of correct responses in 36 or 54 trials vs. chance). We were primarily interested in whether training-induced perceptual gain would transfer to an untrained pair of odor enantiomers or to the untrained nostril. To this end, we performed a series of paired-samples t tests to compare the chiral discrimination accuracies between posttest and pretest for each pair of odor enantiomers presented to the trained nostril as well as to the untrained nostril.

In Experiment 2, as the participants showed significantly improved chiral discrimination for the untrained enantiomer pair that was structurally similar to the trained pair when it was presented to the trained nostril, we performed follow-up Wilcoxon signed rank tests (non-parametric, suitable for small sample sizes) to specifically examine whether those trained with the enantiomers of carvone (n = 6) also demonstrated improved discrimination (posttest vs. pretest) for the enantiomers of limonene and α-pinene, and whether those trained with the enantiomers of limonene (n = 6) also demonstrated improved discrimination for the enantiomers of carvone and α-pinene. In addition, we performed paired-samples t tests to compare the increments in chiral discrimination accuracy (1) between the trained enantiomer pair and the untrained but structurally similar pair presented to the trained nostril and (2) between the trained enantiomer pair presented to the trained nostril and that presented to the untrained nostril. We also compared in a series of paired-samples t tests the participants' intensity ratings between posttest and pretest for each pair of odor enantiomers (ratings averaged between the two enantiomers in each pair) presented to the trained or the untrained nostril. All statistical tests other than the binomial tests were two-tailed.

## Acknowledgments

We thank Zhonghua Lu for comments and suggestions. This work was supported by the Key Research Program of Frontier Sciences (QYZDB-SSW-SMC055) and the Strategic Priority Research Program (XDB32010200) of the Chinese Academy of Sciences, the National Natural Science Foundation of China (31830037 and 31422023) and Beijing Municipal Science and Technology Commission.

## Additional information

### Funding

| Funder | Grant reference number | Author |
|---|---|---|
| Chinese Academy of Sciences | Key Research Program of Frontier Sciences, QYZDB-SSW-SMC055 | Wen Zhou |
| Chinese Academy of Sciences | Strategic Priority Research Program, XDB32010200 | Wen Zhou |
| National Natural Science Foundation of China | 31422023 | Wen Zhou |
| National Natural Science Foundation of China | 31830037 | Wen Zhou |

The funders had no role in study design, data collection and interpretation, or the decision to submit the work for publication.

## Author contributions

Guo Feng, Data curation, Formal analysis, Validation, Investigation, Writing—original draft; Wen Zhou, Conceptualization, Supervision, Funding acquisition, Methodology, Writing—original draft, Writing—review and editing

## Author ORCIDs

Wen Zhou (iD) http://orcid.org/0000-0001-6730-2116

## Ethics

Human subjects: Written informed consent and consent to publish was obtained from participants in accordance with ethical standards of the Declaration of Helsinki (1964). The study was approved by the Institutional Review Board at Institute of Psychology, Chinese Academy of Sciences (H15004).

## Decision letter and Author response

Decision letter https://doi.org/10.7554/eLife.41296.011
Author response https://doi.org/10.7554/eLife.41296.012

# Additional files

## Supplementary files

• Transparent reporting form
DOI: https://doi.org/10.7554/eLife.41296.009

## Data availability

All data analysed during this study are included in the manuscript and supporting files. Source data files have been provided for Figures 1 and 2.

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
