## [Decision Letter]

Thank you for submitting your article "Nostril-specific and structure-based olfactory learning of chiral discrimination in human adults" for consideration by *eLife*. Your article has been reviewed by two peer reviewers, and the evaluation has been overseen by a Reviewing Editor and Andrew King as the Senior Editor. The following individuals involved in review of your submission have agreed to reveal their identity: Rafi Haddad (Reviewer #1); Joel Mainland (Reviewer #2).

The reviewers have discussed the reviews with one another and the Reviewing Editor has drafted this decision to help you prepare a revised submission.

Summary:

This study addressed the question of whether perceptual learning occurs at a peripheral level or in the central nervous system in an olfactory discrimination task. The novelty of this study is twofold: (1) the authors demonstrate the nostril specificity of odor discrimination ability, and (2) the authors provide insights into the specific aspects of stimuli that the subjects learn. That is, perceptual leaning generalizes to other odorant pairs that share common structural configuration. Together, the data support the idea that a certain type of learning to discriminate two odors can happen in the periphery.

Essential revisions:

Although the findings are interesting, both reviewers raised the concern that the authors have made misleading interpretations of their findings. Below the two reviewers each point out this problem in a slightly different ways. We would like the authors to revise their manuscript to more carefully draw their conclusions more in line with the data.

*Reviewer #1:*

General assessment:

The results presented in this manuscript are interesting and important. The experiments are straight forward (but see list of issues below). Overall I like the study. My main reservation is related to positioning and interpreting the result.

Main:

The authors position the results as if it contrast previous studies in the way odor information is shared across nostrils: "contrary to earlier beliefs, one nostril does not readily know what the other learns". However, the Mainland study is based on a detection task and the current study is a discrimination task. This alone can explain the different results.

In my opinion, the interesting part here is that odor discrimination tasks may involve plasticity that is neither transferred to, nor accessible from the untrained side. This is in line with results in other sensory systems (e.g. Karni and Sagi). How this result can be reconciled with the Mainland result should be discussed in the Discussion.

In my opinion, it is also important to discuss the results in light of what we know from rodent experiments on bilateral odor processing. The Kucharski and Hall (1987) studies showed that odor association can be unilateral but it could be accessed bilaterally. The failure to discriminate enantiomers from the untrained nostril reported in this study could be explained by assuming that discrimination is achieved by learning to pay attention to weakly responding M/T neurons that respond differently to the two enantiomers. Since only a subset of M/T cells are inter-connected and these connections are weak (Grobman et al., 2018) the weakly activated M/T cells cannot activate their mirror-symmetric M/T cells resulting in a unilateral odor plasticity. Other explanations that involve plasticity in the piriform cortex can be suggested. In any case, the manuscript should discuss their result in light of these important previous results.

Technical important issues:

1) It is now customary to present in the graphs all data points and not only the average and standard deviation (e.g., Figure 1C-D). This provides a better understanding of the data variability. Please show all data points and show the results for each of the two odor pairs.

2) The Materials and methods sections must be extended and improved as essential parts are not well explained.

a) It is not clear to me how exactly the subjects performed the task (how the bottle looked and handled, did they used new bottles/odors every day or the same bottles for the entire experiment)?

b) Was the same bottle used for the posttest with the trained nostrils also used with the untrained nostril?

c) What statistical tests were used? Sometimes it is t-test (I guess from the reported t value) and sometimes Wilcoxon signed test. Please state which test is used and provide justification for the selection of each test.

d) In this regard: Was the analysis of comparing discrimination success rates done against the expected 33% or against the mean of the pretest group? It may give different results, for example the result shown in the right panel of Figure 2C might suggest that there is a small but significant improvement in the untrained nostril (P = 0.059 which is probably < 0.05 if you compare to 33% expected chance level). While this is not a dramatic change from the paper statement it might show that some information is transferred.

3) Why do the posttest results look significantly lower than the average of the last day in both experiments (Figures 1C and 2B)?

*Reviewer #2:*

This manuscript describes a straightforward experiment that addresses an important issue: is learned odor discrimination achieved at the level of the olfactory mucosa or in the CNS? The described experiments convincingly demonstrate that learning to discriminate two odors can happen in the periphery.

The only major concern is that the authors have an overly broad interpretation of their findings at two points.

First, the current experiments do not definitively show that the nostril is learning molecular features rather than odor quality. Just because the racemic mixtures can be discriminated and are rated as dissimilar doesn't mean that the subjects aren't learning a sub-quality of the odors that allows them to perform the discrimination.

Second, the statement that "human olfactory perception is a learned trait" is a very broad conclusion from these results. This manuscript does not rule out the possibility that some of olfactory perception is innate.

[Editors' note: further revisions were requested prior to acceptance, as described below.]

Thank you for resubmitting your work entitled "Nostril-specific and structure-based olfactory learning of chiral discrimination in human adults" for further consideration by *eLife*. Your revised article has been favourably evaluated by two peer reviewers, and the evaluation has been overseen by a Reviewing Editor and Andrew King as the Senior Editor. The following individuals involved in review of your submission have agreed to reveal their identity: Rafi Haddad (Reviewer #1); Joel Mainland (Reviewer #2).

The manuscript has been improved but there are some remaining issues that need to be addressed before acceptance. Specifically, both reviewers thought that you have addressed their previous concerns very well, and that the manuscript warrants publication at *eLife*. As you will see below, however, one of the reviewers still raised a concern that the claim about generalization based on molecular features remains inconclusive. We think that this can be addressed by revising the text. We would therefore like to invite you to resubmit a revised version of the manuscript.

Essential revisions:

The current experiments still do not definitively show that the nostril is generalizing based on molecular features rather than odor quality. To illustrate this, thegoodscentscompany.com describes (-) carvone as "sweet spearmint herbal minty" and (+)-limonene as "citrus orange fresh sweet".

An alternative explanation to the authors' molecular feature theory is that they are training subjects to discriminate sweet from non-sweet. This presumably doesn't directly relate to structure as the "sweet" odor is a (+) enantiomer for one odor and (-) enantiomer for another. Instead, it is a sub-quality that emerges from the whole set of receptors being activated. Neither enantiomer of pinene is "sweet", so the learning doesn't generalize. This is not the strongest argument, but the current experiments do not rule it out. I suggest the authors soften their language and suggest that there may be alternative explanations. The statement that the "learned sub-quality had to be a direct derivative of the enantioselectivity", for example, is incorrect.

---

## [Author Response]

Reviewer #1:Main:The authors position the results as if it contrast previous studies in the way odor information is shared across nostrils: "contrary to earlier beliefs, one nostril does not readily know what the other learns". However, the Mainland study is based on a detection task and the current study is a discrimination task. This alone can explain the different results.In my opinion, the interesting part here is that odor discrimination tasks may involve plasticity that is neither transferred to, nor accessible from the untrained side. This is in line with results in other sensory systems (e.g. Karni and Sagi). How this result can be reconciled with the Mainland result should be discussed in the Discussion.

The reverse hierarchy theory (Ahissar and Hochstein, 2004) nicely reconciles our results with the internarial transfer of learning of androstenone detection observed in the Mainland et al. study (Mainland et al., 2002). Supposing that learning proceeds as a countercurrent along the olfactory processing hierarchy, the detection of androstenone could be achieved by heightened attention or sensitization to its olfactory quality (amplification of a weak signal) and hence possibly engages a higher generalizing level, whereas the learning of chiral discrimination likely relies more on lower-level inputs and is thus more specific. These are now incorporated in the second paragraph of the Discussion section.

In my opinion, it is also important to discuss the results in light of what we know from rodent experiments on bilateral odor processing. The Kucharski and Hall (1987) studies showed that odor association can be unilateral but it could be accessed bilaterally. The failure to discriminate enantiomers from the untrained nostril reported in this study could be explained by assuming that discrimination is achieved by learning to pay attention to weakly responding M/T neurons that respond differently to the two enantiomers. Since only a subset of M/T cells are inter-connected and these connections are weak (Grobman et al., 2018) the weakly activated M/T cells cannot activate their mirror-symmetric M/T cells resulting in a unilateral odor plasticity. Other explanations that involve plasticity in the piriform cortex can be suggested. In any case, the manuscript should discuss their result in light of these important previous results.

We thank the reviewer for directing us to the helpful references and for the explanation regarding why both bilateral access to unilaterally stored olfactory associations and unilateral odor plasticity (as observed in the current study) are possible. They are now incorporated in the first paragraph of the Discussion section.

Technical important issues:1) It is now customary to present in the graphs all data points and not only the average and standard deviation (e.g., Figure 1C-D). This provides a better understanding of the data variability. Please show all data points and show the results for each of the two odor pairs.

We have plotted all data points for all enantiomer pairs in Figure 1—figure supplement 1 and Figure 2—figure supplement 1.

2) The Materials and methods sections must be extended and improved as essential parts are not well explained.a) It is not clear to me how exactly the subjects performed the task (how the bottle looked and handled, did they used new bottles/odors every day or the same bottles for the entire experiment)?

The odor enantiomers were presented in identical 280ml narrow-mouthed glass bottles. Each bottle contained 10 ml clear liquid and was connected with a Teflon nosepiece. The participants were blindfolded during pretest, training and posttest and were instructed not to grasp or touch the bottles. In each trial of the triangular unirhinal odor discrimination task, the experimenter presented the bottles to the participant’s open nostril (positioning the nosepieces into the open nostril), one at a time. The odor solution in each bottle was replaced with a freshly prepared solution of the same compound every two days. These details are now clarified in the Materials and methods section of the revised manuscript.

b) Was the same bottle used for the posttest with the trained nostrils also used with the untrained nostril?

Yes. We have made it explicit in the Materials and methods subsection “Procedure”.

c) What statistical tests were used? Sometimes it is t-test (I guess from the reported t value) and sometimes Wilcoxon signed test. Please state which test is used and provide justification for the selection of each test.

We have now described the statistical analyses in detail in the Materials and methods subsection “Analysis”.

d) In this regard: Was the analysis of comparing discrimination success rates done against the expected 33% or against the mean of the pretest group? It may give different results, for example the result shown in the right panel of Figure 2C might suggest that there is a small but significant improvement in the untrained nostril (P = 0.059 which is probably < 0.05 if you compare to 33% expected chance level). While this is not a dramatic change from the paper statement it might show that some information is transferred.

The comparison was between posttest and pretest since we were primarily interested in whether training-induced perceptual gain would transfer to an untrained pair of odor enantiomers or to the untrained nostril. We have made this explicit in the Materials and methods subsection “Analysis”.

3) Why do the posttest results look significantly lower than the average of the last day in both experiments (Figures 1C and 2B)?

As can be seen from Figure 1—figure supplement 1 and Figure 2—figure supplement 1 (left panels, trained nostril), this pattern was mainly driven by a minority of the participants whose posttest discrimination accuracies for the trained pair of odor enantiomers fell below 60% (2 in Experiment 1 and 3 in Experiment 2). As the posttest was conducted an hour after the completion of the last training session, the participants could be fatigued. It was also possible that the lack of feedback at the posttest hampered their performance.

Reviewer #2:[…] The only major concern is that the authors have an overly broad interpretation of their findings at two points.First, the current experiments do not definitively show that the nostril is learning molecular features rather than odor quality. Just because the racemic mixtures can be discriminated and are rated as dissimilar doesn't mean that the subjects aren't learning a sub-quality of the odors that allows them to perform the discrimination.

The discrimination of odor enantiomers ultimately draws upon the enantioselectivity of olfactory receptors. If what was learned was a sub-quality that enabled the discrimination between the (+) and (-) forms of both carvone and limonene but not those of α-pinene, this sub-quality had to be a direct derivative of the enantioselectivity of the olfactory receptors shared by carvone and limonene. In this sense, the learning of chiral configurations and that of odor sub-qualities contingent on chiral configurations are flip sides of the same coin.

In the last paragraph of the Results section, we have clarified that our results do not exclude the possibility that the participants performed the chiral discriminations based on a learned sub-quality of the odors. Rather, they argue that this learned sub-quality had to be a direct derivative of the enantioselectivity of the olfactory receptors shared by carvone and limonene.

Second, the statement that "human olfactory perception is a learned trait" is a very broad conclusion from these results. This manuscript does not rule out the possibility that some of olfactory perception is innate.

We fully agree and have changed the statement to “human olfactory perception can partially be a learned trait”.

[Editors' note: further revisions were requested prior to acceptance, as described below.]

Specifically, both reviewers thought that you have addressed their previous concerns very well, and that the manuscript warrants publication at eLife. As you will see below, however, one of the reviewers still raised a concern that the claim about generalization based on molecular features remains inconclusive. We think that this can be addressed by revising the text. We would therefore like to invite you to resubmit a revised version of the manuscript.The current experiments still do not definitively show that the nostril is generalizing based on molecular features rather than odor quality. To illustrate this, thegoodscentscompany.com describes (-) carvone as "sweet spearmint herbal minty" and (+)-limonene as "citrus orange fresh sweet".An alternative explanation to the authors' molecular feature theory is that they are training subjects to discriminate sweet from non-sweet. This presumably doesn't directly relate to structure as the "sweet" odor is a (+) enantiomer for one odor and (-) enantiomer for another. Instead, it is a sub-quality that emerges from the whole set of receptors being activated. Neither enantiomer of pinene is "sweet", so the learning doesn't generalize. This is not the strongest argument, but the current experiments do not rule it out. I suggest the authors soften their language and suggest that there may be alternative explanations. The statement that the "learned sub-quality had to be a direct derivative of the enantioselectivity", for example, is incorrect.

We have followed the reviewer’s suggestion and removed the statement that the “learned sub-quality had to be a direct derivative of the enantioselectivity of the olfactory receptors shared by carvone and limonene”. We have also acknowledged that our data “did not fully exclude the possibility that the participants performed the chiral discriminations based on a learned sub-quality of the odors”. As an aside, we would like to note that in terms of absolute configuration, both (-) carvone and (+) limonene have a rectus configuration.